# Artificial web of disclination lines in nematic liquid crystals

Mengfei Wang[1], Yannian Li[1] & Hiroshi Yokoyama [1]

Disclinations are topological singularities of molecular arrangement in liquid crystals, which typically occur when the average orientation of molecules makes a $\pi$ rotation along a fictitious closed loop taken inside the liquid crystal. Depending on the sense of molecular rotation, the disclination lines are either of 1/2 or −1/2 strength. When two disclination lines with the opposite strength meet, they are annihilated without trace. It is hence generally considered difficult in the nematic phase to stabilize a condensed array of free-standing disclination lines without the aid of topological objects like colloidal inclusions. Here we show that a free-standing web of 1/2-strength twist disclination lines can be stably formed in thin liquid crystal cells by means of a judicious combination of orientationally patterned confining surfaces fabricated by the micropatterned photoalignment technique. Theoretical model indicates that disclination lines are held apart at the intersection by a repulsive force generated by the Frank elasticity.

[1] Liquid Crystal Institute, Kent State University, 1425 Lefton Esplanade, Kent, OH 44242, USA. Correspondence and requests for materials should be addressed to H.Y. (email: hyokoyam@kent.edu)

A s topological structures associated with molecular orientation in liquid crystals (LCs), the disclinations, existing in the form of lines and points, manifest the microscopic symmetry of the LCs, thereby serving as a structural signature of particular archetype of LCs[1, 2]. The nematic LC, exclusively used in liquid crystal display TVs today, was in fact named after the Greek word for thread, i.e., the disclination line, ubiquitously encountered in this type of LCs[1, 2]. Of all the known LC phases, the nematic LCs possess the highest symmetry, characterized by the nonpolar uniaxial orientational order of the constituent rod-shaped molecules, whereas there is no long-range positional order as in the ordinary liquid, allowing for easy flows with a relatively low viscosity. Although the constituent molecules are often polar, they preferentially orient along the axis of the uniaxial symmetry without discriminating the head from the tail of the molecule. The direction of the axis of uniaxial symmetry is mathematically specified by a unit vector called the director **n**. States with **n** are locally equivalent to the state with −**n** due to the nonpolarity of the nematic phase. The ground state of the nematic LC is the uniform state with the spatially invariant **n**.

Disclination lines in nematic LCs appear when the director makes a continuous π rotation along a closed loop, converting **n** to −**n**. Because of the physical equivalence of **n** and −**n**, there is no discontinuity of the LC structure except along the line singularity running somewhere inside the loop. Depending on the sense of rotation of the director along the loop, there arise two subtypes referred to as 1/2 and −1/2-strength disclinations, although they are not topologically distinct[3]. This type of disclination lines are densely formed at the phase transition, when the isotropic liquid is suddenly cooled down to the nematic phase, because of the nucleation of numerous nematic LC domains with random director orientations. The disclination lines thus formed, however, rapidly disappear through pair annihilation and loop shrinkage for the aforementioned equivalence of 1/2 and −1/2 disclinations[3]. The time evolution of disclination lines at the phase transition has intrigued even cosmologists as a laboratory model of the early universe[4], based on the mathematical equivalence of fundamental equations governing these two entirely different systems. For the past decade, there have been attempts for artificial manipulation of disclination lines by means of topologically conflicted boundaries[5–7] and by use of topological objects like colloidal inclusions[8, 9]. Nevertheless, their inherent readiness for annihilation has hampered the realization of condensed array of free-standing disclination lines in nematic LCs as those involving line intersectoins.

We describe here a scheme to design and fabricate a free-standing network of twist disclination lines without the aid of colloidal particles. The key ingredient is the substrates with a microscopically patterned surface alignment for LC molecules that not only nucleate twist disclination lines but also exert elastically mediated forces on the disclination lines in such a way as to stabilize intersecting disclination lines. In the next section, we present the theoretical expressions of the forces in play in this system, followed by the experimental demonstration that a free-standing web of 1/2-strength disclination lines can be stably formed in thin LC cells through a proper combination of orientationally patterned confining surfaces. According to the theoretical estimates of the forces acting between disclination lines and

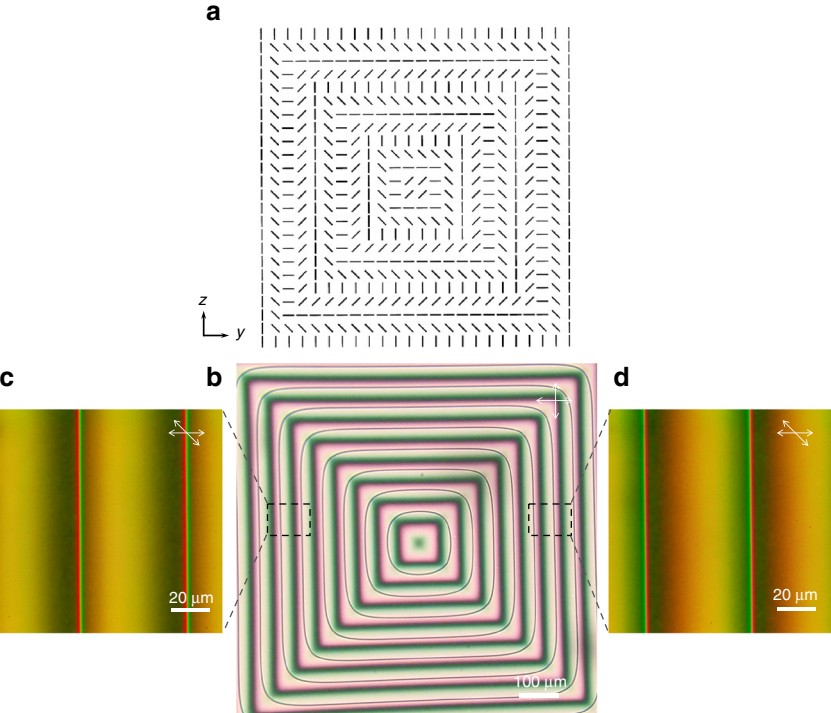

**Fig. 1** Square surface alignment pattern and the array of 1/2-strength twist disclination lines. **a** The designed square pattern for surface photoalignment of nematic liquid crystal consisting of periodic and concentric orientational variations for the azimuthal anchoring direction. **b** Polarized micrograph of a liquid crystal cell held between the square patterned substrate and a uniformly surface-aligned substrate. For every pitch (53 μm) of in-plane rotation of the director, a free-standing loop of twist disclination line is generated (seen as concentric loops of *thin dark lines*). Except for the corner, the disclination lines follow the path where the alignment on the patterned surface is orthogonal to the uniform alignment on the opposite surface. **c**, **d** Magnified views of **b** between obliquely oriented polarizers to indicate the continuous and asymmetric variation of the twist states across the disclination lines that appear as *sharp colored lines* in the figure. The disclination lines in **c**, **d** are of the opposite handedness; in **c**, the *left* (*right*) side of the disclination is associated with −π/2 (π/2) twist from the *bottom* to the *top surface*, corresponding to the plus sign in Eq. (1) and hence is of the right handedness. That in **d** is of the left handedness. The oppositely twisted states across the disclination line give an asymmetric contrast under obliquely aligned polarizers

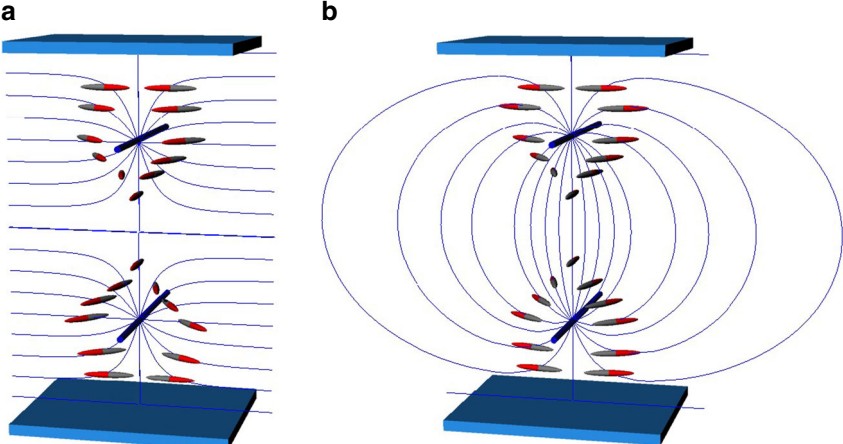

**Fig. 2** Director configurations around two parallel twist disclination lines confined in a planar cell. **a** Configuration depicted according to Eq. (1) for the same handedness case. **b** Configuration of the opposite handedness case. Although the director of a nematic LC does not have a head–tail asymmetry, different colors are applied to highlight the differences in the topology of twist states. In **a**, the pair of the disclination lines separate the regions of $\pm \pi$ twisted states; in **b**, both sides of the disclination pair are untwisted, and the deformation is confined in the vicinity of the disclinations, which creates the attractive force between the disclination lines. The configuration in **b** is not in mechanical equilibrium without the aid of external force

between the patterned surface and the disclination line, twist disclination lines are prevented from annihilation through the balance of the elastic forces over a wide range of intersecting angles. In this regard, the present disclination web can be regarded as a two-dimensional analog of the cholesteric blue phase[10, 11].

## Results

**Orientationally patterned substrates**. We use a topological surface orientation pattern (2.6 mm × 2.6 mm) as shown in Fig. 1a, which possesses a two-dimensional chirality, and is strongly incompatible with the uniform orientation of the LC director **n** (the unit vector along the average orientational axis of molecules). Put into contact with this pattern, the LC director is forced to continuously wind in the plane from the center of the pattern toward the edge. The LC tries to resolve the structural incompatibility by periodically introducing a 1/2-strength twist disclination line for every $\pi$ rotation of the director, which leads to the generation of an array of closed disclination loops concentric with the surface pattern with small rounding at the corner (Fig. 1b–d), which is caused by the balance between the line tension of the disclination and the elastic stress generated by the imbalance of the twist deformations across the disclination line (for detailed discussions, see Supplementary Notes 1 and 2). This situation is similar to cholesteric LCs confined in a Cano-wedge[1, 12] that is made of two planar aligned substrates held together at a small angle to induce linear variation of thickness. The role played by the thickness in the Cano-wedge is now taken by the continuous rotation of the director on the surface pattern. Note that the disclination lines are of twist type (the director being everywhere parallel to the surface) separating regions of quantized twist angles by the unit of $\pi$ due to the equivalence of **n** and −**n**.

**Theoretical analysis of forces on disclination lines**. For a given boundary condition at the bounding surface, the equilibrium distribution of the director can be theoretically found as the one that minimizes the Frank elastic free energy, written as a sum of contributions from the three independent mode of deformations referred to as the splay, twist, and bend modes[1]. These modes are associated with specific elastic constants, $K_{11}$, $K_{22}$, and $K_{33}$, respectively. Here, we give only the salient results of the analysis,

leaving the detailed derivation to Supplementary Note 1. Under a fairly general condition of equal splay and bend Frank elastic constants, i.e., $K_{11} = K_{33} \equiv K$, it is shown that the equilibrium director profile is governed by a linear differential equation for the azimuthal angle of the director $\varphi$. The linearity allows superposition of solutions, making it possible to decompose the problem into simpler sub-problems. We then first consider a uniform $\pi/2$-twisted cell, and consider the effect of surface pattern later. The director profile around a straight 1/2-strength twist disclination line in a $\pi/2$-twisted cell of thickness $L$ can be analytically solved as[13–15]

$$\varphi = \pm \frac{1}{2} \left[ \tan^{-1} \left( \frac{\sin \frac{\pi x}{L} \cosh \sqrt{\frac{K_{22}}{K}} \frac{\pi y}{L} - \sin \frac{\pi d}{L}}{\cos \frac{\pi x}{L} \sinh \sqrt{\frac{K_{22}}{K}} \frac{\pi y}{L}} \right) + \frac{\pi}{2} \operatorname{sgn}(y) \right],$$

$$(1)$$

where the $x$-axis is perpendicular to the cell surfaces located at $x = L/2$, $−L/2$, and the $z$-axis is taken along the disclination line running at $x = d$, $y = 0$. The plus–minus sign indicates the right and left handedness of the disclination line, which becomes nonequivalent when the entire system is chiral as in the present case. Along the $x$-axis right below and above the disclination, $\varphi$ remains constant, but makes a $\pi/2$ jump across the disclination line. The induced distortions around the disclination line relax into the simple $\pi/2$-twist over the distance of $L\sqrt{K/K_{22}}$.

The line tension associated with the disclination is given by

$$\gamma = \frac{\pi}{4} \sqrt{KK_{22}} \ln \left( \frac{L}{\pi \delta_c \cos \frac{\pi d}{L}} \right) + f_c,$$

$$(2)$$

where $f_c$ and $\delta_c \sim 10$ nm are the energy and the radius of the disclination core[16]. As this formula shows, the disclination is most stable at the middle of the twist cell, and as it approaches the boundary, there emerges an increasing repulsive force due to the growing twist deformations[17].

The same analysis can be extended to the case of two parallel disclination lines located at $x = d_1$, $d_2$, $y = 0$, yielding an additional interaction energy between disclinations as

$$\gamma_{\pm} = \pm \frac{\pi}{4} \sqrt{KK_{22}} \ln \left( \sin \frac{\pi d_1}{L} - \sin \frac{\pi d_2}{L} \right)^{-2},$$

$$(3)$$

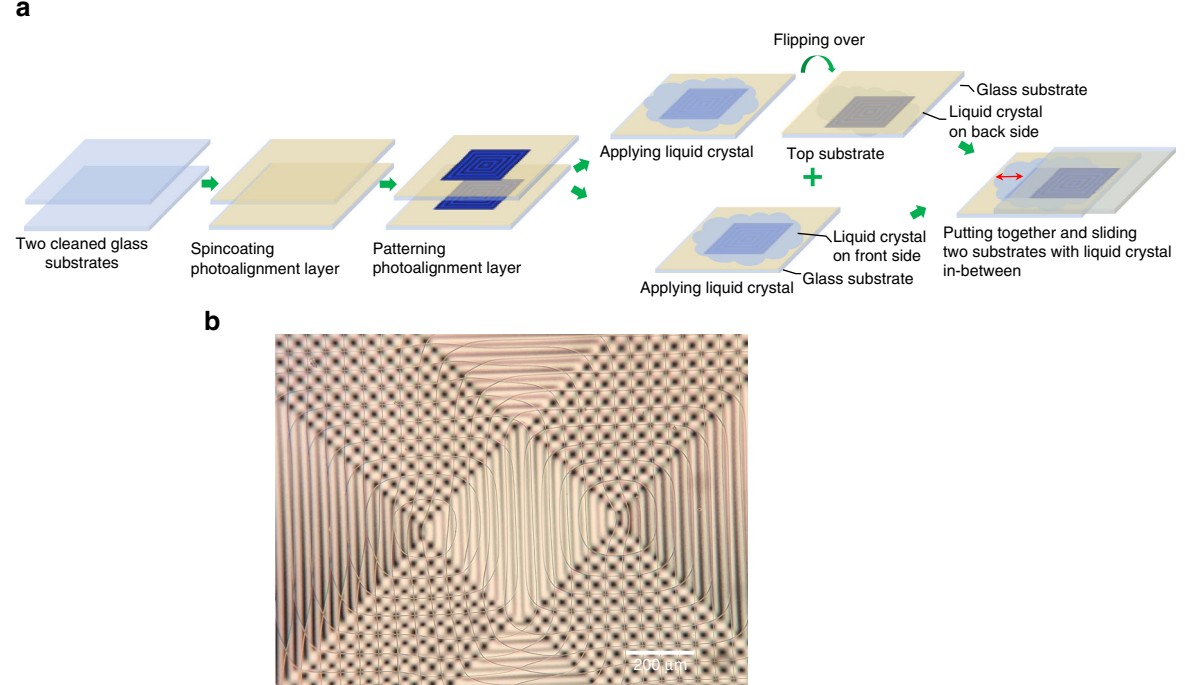

**Fig. 3** Preparation steps and the polarized micrograph of assembled sliding cell. **a** Preparation steps of patterned photoalignment cell with sliding substrates. **b** Polarzied optical micrograph of the assembled cell immediately after the top substrate was slid to form the desired configuration of overlapped photoalignment patterns. The lack of coloration indicates the thickness of the liquid crystal layer is still larger than 30 μm. The disclination lines in the central *rectangle* are still intact, which begin to annihilate as the LC layer thins. The disclination lines are more susceptible to bending in a thick LC layer than at later stages with a thinner LC layer, generally due to the weakening of some of the stabilizing forces with the increase in the thickness

where the plus and minus signs apply to the same and the opposite handedness of the disclination lines, indicating that the disclination lines repel each other when they are of the same handedness. It must be noted that the twist angles on both sides of disclinations are $\pm\pi$ or 0 depending on the same and the opposite handedness of the disclinations, so that the cell is of a uniform planar structure. Combining the repulsion from the boundary, one finds that the two disclination lines of the same handedness are located at $x = -L/4, L/4$ in equilibrium, while those with the opposite handedness attract each other, merge and disappear as illustrated in Fig. 2. Even when the parallel disclinations are laterally separated in the $y$ direction, it is shown that the disclination lines of same (opposite) handedness repel (attract) each other. When, in particular, $d_1 = d_2$ and the lateral separation $h$ is smaller than the cell thickness, the lateral force, both repulsive and attractive, is proportional to $1/h$ and is independent of the vertical position (for more details, see Supplementary Note 1).

The extra twist energy stored around the disclination lines is responsible for the repulsive and attractive forces. When a pair of straight disclination lines are intersecting at an oblique angle, the force between the disclination lines is localized in the vicinity of the intersection over the range of the cell thickness. Extending the analysis to the cases of obliquely oriented (non-parallel) straight disclination lines, we can show that the force between disclination lines (in the area of intersection) continuously varies between repulsive to attractive regime, proportional to $\cos\Phi$ with $\Phi$ being the angle between the disclination lines when projected to the surface. As one of the disclination lines of the same handedness is rotated in-plane relative to the other (with the boundary fixed), the repulsive force at the intersection gradually decreases and vanishes when the lines are orthogonal. Further rotation leads to the increase of attractive force; in other words, the handedness of the disclination is switched to the opposite at $\Phi = \pi/2$ (Supplementary Note 1).

The last component of force on disclination lines comes from the patterned surface orientation itself. The square pattern locally consists of a continuous one dimensional change of the azimuthal angle as

$$\varphi_s = \frac{\pi}{p} y, \qquad (4)$$

where $p$ is the pitch of the pattern for $\pi$ rotation of the director. On such a surface, disclination lines of one handedness become more favorable than the other. On the patterned substrate as described by Eq. (4), a right-handed disclination line is subjected to a pitch-dependent force pulling the disclination down toward the substrate:

$$f_P = -\frac{\pi^2}{2p} K \left(1 - 2\frac{d}{L}\right). \qquad (5)$$

Disclinations of the opposite handedness will be pushed up by the upward force of the same magnitude. If the top substrate is patterned like Eq. (4), the force contribution from the top surface adds up to Eq. (5) to yield a position-independent force as

$$f_P = -\frac{\pi^2}{p} K. \qquad (6)$$

The balance of these forces determines the distribution of the disclination lines and their stability in three dimensions.

**Web of twist disclination lines**. We assembled a cell using a pair of square patterned substrates, each bearing an array of disclination loops. The loops are floating in the cell without contact with the surfaces. The orientational pattern consisting of $p = 53$ μm concentric periodic bands was fabricated by the photoalignment patterning technique using the polarized micro-projection system reported elsewhere[18, 19]. The local alignment direction is defined perpendicular to the linear polarization of the irradiated

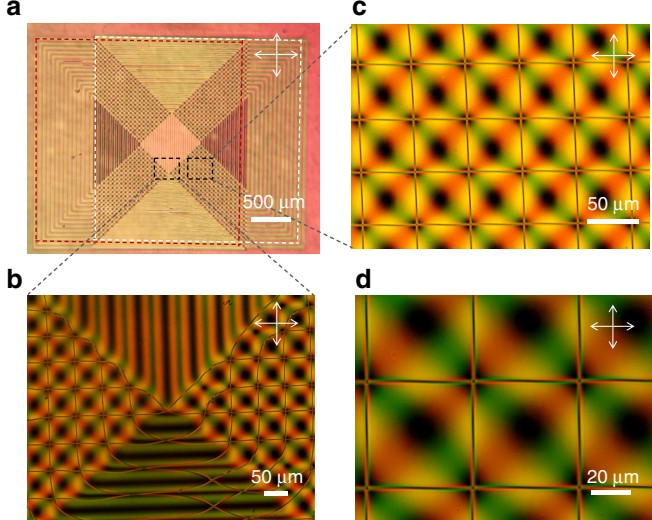

**Fig. 4** Two-dimensional web of 1/2-strength twist disclinations induced by the surface orientational pattern. **a** Entire top view of the LC layer between partly overlapped square patterns shown in Fig. 1a (surrounded by *broken red* and *white lines*) observed between crossed polarizers. **b** Magnified view of **a** at the lower corner of the central region. **c**, **d**, Magnified polarized micrographs of **a** showing the square web of disclinations. The color contrast manifests the local twist and birefringence. The lattice spacing of the disclination web is 53 μm, commensurate with the periodicity of surface alignment. Disclination lines that were initially present in the central region were quickly annihilated, leaving a single loop of *diagonal line* surrounding the area. In the lower segment, the stripe patterns from the bottom and top substrates are horizontally parallel where the disclination lines are of the same handedness, assuring their stability. All the web structure is indefinitely stable, once the central region becomes disclination-free. Around the intersection **c**, **d**, the twist angle is 0, π, 0, and −π counter clockwise from the first quadrant. On the diagonal connecting the *left bottom* intersection and the *right top* one, the twist angle is 0, but the optic axis is continuously winding in-plane as manifested by the periodic change of contrast. With the birefringence of ZLI-2293 being Δn ~ 0.13, the interference color gives the estimate of the cell thickness to be ca. 10 μm

ultra violet light. The alignment pattern was first split into segments of identical orientation, and each segment was sequentially irradiated onto a glass substrate coated with a photoaligning azo dyes[20] while adjusting the polarization direction as desired. At saturated irradiation, the azimuthal anchoring energy of the photoaligned surface exceeds $10^{-4}$ m J m$^{-2}$, which can be considered perfectly strong for cells thicker than 1 μm. A room temperature nematic LC (ZLI-2293, Merck) was applied onto the patterned surface. When temporarily covered for observation purposes with a glass slide whose surface was also photoaligned to give a uniform planar alignment, a regularly spaced array of 1/2-strength twist disclination loops was observed under the polarized optical microscope (Olympus) as shown in Fig. 1b. Here, the position of the disclination lines in relation to the base square periodic pattern is dictated by the orientation of the uniform alignment on the cover substrate in such a way that the disclination loop follows the pattern where the surface orientation is perpendicular to the uniform alignment imposed by the counter substrate. At the corners, however, the line tension of the disclination induces a rounding to make a shortcut. The continuous contrast changes between crossed polarizers confirm that the alignment in each unit segment is along the desired orientation. The sense of in-plane winding of director selects a unique handedness for the disclination lines to better accommodate the director profile around the disclination.

Flipping one of the patterned substrates loaded with the nematic LC over the other (without the cover slide), we obtained a LC layer held between the square patterned substrates (Fig. 3a). Starting with the initial state wherein there was no overlap of square patterns, the substrate was manually slid to get a gradually increasing overlap of the patterns. Through the sliding process, it was easy to keep the disclination array intact despite a slight viscous drag. A striking observation is the outstanding stability of the disclination lines even when the two separate disclination arrays began to overlap each other. The thickness of the LC layer could not be precisely controlled, but was always in the range of 10–40 μm (Fig. 3b).

The substrates were slid until an overlap over 2/3 of the square patterned area was reached as shown in Figs. 3b, 4a, and Supplementary Movie 1. Here, all the possible combinations of the alignment patterns are simultaneously observable. At the center of Figs. 3b and 4a is a 45°-tilted rectangular region made by the overlap of the vertical stripes from the same quadrants of the square pattern on the top and bottom surfaces. It must be noted here that the effect of physical flipping of a disclination-bearing substrate is equivalent to $y \rightarrow -y$ in Eq. (1), thereby reversing the handedness of the disclination lines. Only in this central area, the disclination lines were unstable and completely disappeared through pair annihilation, reconnection, and loop shrinkage (Fig. 4b). It is worth pointing out that the disclination lines can be metastable even in this region when the LC layer is sufficiently thick because of the surface force Eq. (6) that traps the line near the substrate. It follows from Eqs. (2), (3), and (6), the threshold thickness is given by $L_{th} = \frac{1}{2}p\sqrt{K_{22}/K}$, below which the disclination lines of opposite handedness are always unstable. For $p = 50$ μm, one roughly finds $L_{th} \sim 15$ μm, which is consistent with the present observations.

The disclination-free region is surrounded by four areas where the stripes on the bottom and the top surfaces are orthogonal. The disclination lines, put forward from the surface patterns, form a web of disclinations comprised of regular square lattice commensurate with the underlying surface patterns (Fig. 4c, d). The web remains indefinitely stable in the ca. 10 μm-thick cell. The polarized microscopy indicates that the lattice points are pinned at the points where the local directions of surface alignment on the bottom and the top substrates are parallel (Fig. 5a). The adjacent quadrants of the lattice point are of 0, π, 0, and −π twisted states counter clockwise from the first quadrant (Fig. 5b, c; Supplementary Note 1). More quantitatively, the twist angles in this area are distributed according to $\varphi_t = \pm \pi y/p \pm \pi z/p$, where the plus and minus signs refer to which sides of the square have been put together. The bottom right region magnified in Fig. 3b is described by $\varphi_t = \pi y/p - \pi z/p$, which indicates that the twist angle is constant along the diagonal axes from the bottom left to the top right. At the corners and the middle points of the square lattice, the disclination lines are energetically anchored on the diagonal lines on which $\varphi_t = 0$, $\pm \pi$, and $\varphi_t = \pm \pi/2$, respectively. At other points, the straight disclination lines are not in mechanical equilibrium with the unbalanced twist deformations on both sides of the line; however, the line tension of the disclination of about 100 pN is large enough to keep the line virtually straight for short pitches below ca. $p = 50$ μm for the present range of cell thickness. For larger pitch patterns beyond $p = 100$ μm, indeed, we observed a periodical bending of the disclinations (for more detailed discussion, see Supplementary Note 1).

It is energetically more favorable for disclination lines to run periodically along the diagonal axes with $\pm \pi/2$ twists than shaping the square web, since the length of disclination line is reduced by a factor $1/\sqrt{2}$ and the super-twisted states near the $\pm \pi$ twisted quadrants in the square web are converted to the

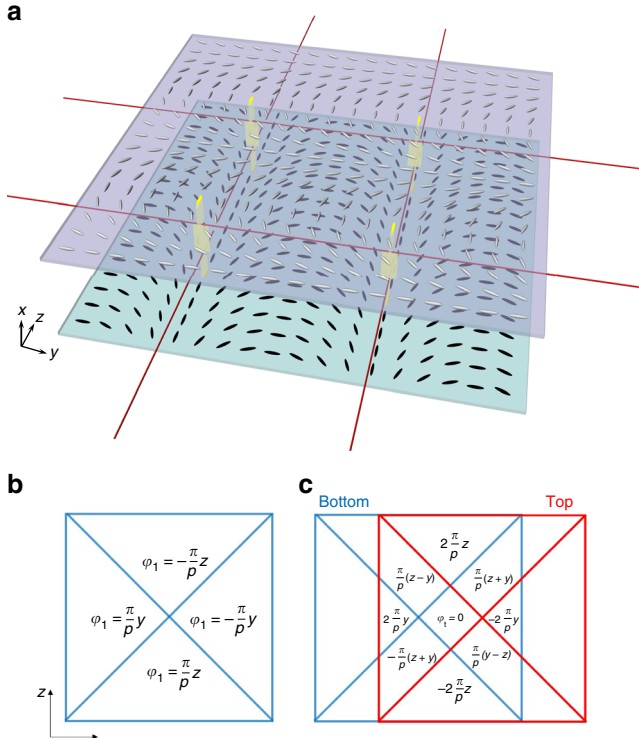

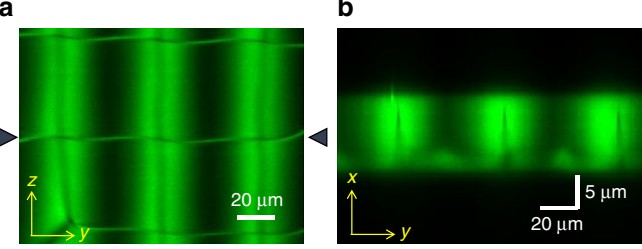

**Fig. 6** Confocal fluorescent microscopy of the square web of twist disclinations in the surface patterned nematic LC cell. An anisotropic fluorescent dye was mixed in the LC. **a** Top view (*y–z* plane) of the cell under a linearly polarized excitation. The disclinations are captured as the *thin horizontal lines* and the *dark lines* inside the broad *bright bands*. **b** Cross-sectional view (*x–y* plane at the center of the cell) along the horizontal disclination line at the center (indicated by *solid triangles* in **a**). The vertical disclination line is located at the tip of the *vertical dark shadow* inside the bright band, and the bright structure running at the lower boundary indicates the *horizontal line*. The spacing between the *vertical* and *horizontal lines* is ca. 5 μm

**Fig. 5** Director profile in the surface patterned cell in the area of square web of disclinations. **a** Schematic illustration of square web of disclinations in a nematic LC cell held between the patterned substrates. Periodic arrays of twist disclinations (*red lines*) are generated by the periodic orientational patterns on the top and bottom substrates (the two *lines* along the *y*-axis are generated from top surface, while the *lines* along *z*-axis are produced by surface alignment from bottom substrate), and are stably located by elastic forces a few micrometers apart from one another at the intersections. The intersections of the disclination lines, highlighted by eye-guiding *yellow strips* giving the sheets of directors, are anchored where the directors on the bottom and the top surfaces are parallel, whereas the middle points of the line are positioned at the place of $\pm\pi/2$ twist. **b** Position dependence of the azimuthal orientation of LC director on the square patterned substrate. Flipping this substrate over to have the top substrate, we see that the azimuthal angle changes its sign irrespective of the manner it is flipped, i.e., $\varphi_2 = -\varphi_1$. **c** Position dependence of the twist angle, $\varphi_t = \varphi_2 - \varphi_1$, in the assembled liquid crystal cell

normal twisted states with the twist angle $< \pi/2$ (for more details, see Supplementary Note 2). Nevertheless, the square web of disclinations was found to be indefinitely sustainable. Even after a vigorous shearing of the substrates to completely disrupt the disclination lines, the square web of disclinations was recovered within a fraction of a minute (Supplementary Movie 2). The remarkable stability of the web can be understood as resulting from the repulsive forces between disclination lines and the forces between the surface pattern and the disclination line. In particular, the effective repulsive force due to surface patterns, Eqs. (5) and (6), acting between the disclination lines of opposite handedness is considered vital to prevent the lines from coalescing for large angles of crossing $\Phi > \pi/2$, where the direct interaction between opposite disclinations, Eq. (3), becomes increasingly attractive. In consistent with this scenario of stabilization, we observed that the web structure becomes much less stable when the pitch of the surface pattern *p* exceeds 100 μm. As long as the pitch *p* is less than ca. 50 μm, the web of disclination lines could be stably fabricated through the same procedure as described here for virtually any angle of intersection

$\Phi$, forming a wide range of rhomboidal webs (Supplementary Movie 3).

To probe the 3D structure of the intersection of disclination lines, we conducted the laser scanning fluorescence confocal microscopy (Fig. 6). The liquid crystal was mixed with a small concentration of anisotropic fluorescent dye that follows the local orientation of the liquid crystal[21]. The disclinations are observable as thin green or dark lines depending on their orientation. Three disclination lines running along the *z*-axis are present at the tip of thin dark shadows in the middle of broad green bands. About 5 μm below these disclinations exists a horizontal narrow band with lighter contrast indicative of the disclination line (Supplementary Movie 4).

## Discussion

In this work, we have demonstrated the artificial creation of web of free-standing twist disclination lines in nematic LCs by engineering the patterned surface alignment so as to control the forces between disclination lines. The planar configuration of the director, characterized by the azimuthal angle only, allows analytical treatment of the relevant forces, which under a fairly general condition turn out to be additive. Unlike full 3D configurations, involving topological inclusions and requiring more angular variables, the parameter space investigated here is large enough to allow a vast range of design possibilities, yet is small enough to allow analytical and intuitive approaches. The planar structure assumed here is not necessary valid particularly when the lateral deformation is significant over the length scale of the cell thickness and/or a short pitch chiral nematic LC is used instead of achiral nematic LCs[14, 15]. It should, however, be worth mentioning that the out-of-plane orientation of the director can be efficiently suppressed even in these cases by using LCs with a negative dielectric anisotropy under a sufficiently intense electric field.

The disclination lines are finest physical features in LCs having significant impacts on the phase structures and physical properties. Although disclinations have long been considered troubling objects to be avoided in device applications, the unique properties of the orientational singularity would inspire positive roles of disclination lines in LCs, once their deliberate design and fabrication technologies have been established. Just like polymeric nanofibers, disclination lines in LCs carry tensile forces; but due to their structural origin, lacking in any physically distinct constituents, the disclination lines are superelastic and indefinitely

extensible without yield point. If webs of condensed disclination lines can be fabricated by design, it would be foreseeable to develop disclination-integrated objects that exhibit a unique mechanical response to external fields and stimuli such as soft shape-memory[22]. Moreover, the abrupt variation of anisotropic optical properties around disclination lines could also be potentially useful as optical media to manipulate the Pancharatnam–Berry geometric phase[19] toward nanophotonic applications. Finally, it has also been demonstrated that disclination lines can trap a variety of nano- and micro-scopic objects for their higher energy states[5, 23]. It should be of interest to use the engineered disclination webs as template to assemble functional objects with nanometer scale precision, flexibility, and responsivity. It may be particularly intriguing if one can use the disclination web as tracks for cargo transport in small dimensions, much like cytoskeletons[24], taking advantage of the excitable property of LCs. For this purpose, however, it remains to be a challenge to develop a scheme to transfer a cargo from one disclination to another without disrupting the integrity of the disclination lines. The stabilizing forces of disclination webs scale as the inverse of cell thickness and the pitch of the surface pattern. The disclination lines tend to behave more rigidly in smaller scales. Therefore, the finer the structure, the more robust the web becomes, making it an ideal platform for soft nanomachines.

## Methods

**Fabrication of the liquid crystal cells**. Glass substrates were cleaned by ultrasonic bath, followed with isopropyl alcohol (IPA) rinsed, oven dried, and ultra violet ozone cleaned in cleanroom. The sulfonic azo dye (SD-1) was synthesized in-house and used as photoalignment material. The solution was prepared with 1.8% wt. SD-1 in dimethylformamide (DMF), mixed with ultrasonic bath for 30 min at 60 °C. The mixture was uniformly spin-coated onto the cleaned glass substrate and dried on hot plate for 1 min at 60 °C. Photo-patterning process described in the previous work[18, 19].

**Laser scanning fluorescence confocal microscope**. The anisotropic fluorescent dye N,N′-bis(2,5-di-tert-butylphenyl)-3,4,9,10-perylenedicarboximide (BTBP) (from Sigma-Aldrich) was used. The BTBP dye was mixed in methanol with 0.01% wt. Then the BTBP solution was mixed with LC ZLI-2293 by the ratio of 1:1 by weight. To protect photoalignment layer from laser scanning for the probe of fluorescence confocal microscope, an additional layer of reactive mesogen RM257 (from Wilshires Technologies), 2.0% wt. in toluene, was spin-coated on top of the SD-1 layer after photo-patterning, before applied LCs, then cured at 305 nm for 30 s. The trace photo-initiator 2-methyl-4'-(methylthio)-2-morpholinopropiophenone (Sigma-Aldrich) was added to the RM257 solution.

**Data availability**. The data sets generated during and/or analyzed during the current study are available from the corresponding author upon reasonable request.

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

## Acknowledgements

We gratefully acknowledge O.D. Lavrentovich, M. Model, and B. Li for their generous support in the fluorescence confocal microscopy.

## Author contributions

H.Y. conceived the project, carried out the model analysis, designed the experiments, discussed the results, and worked on the manuscript. M.W. designed and carried out the experiments, discussed the results, and worked on the manuscript. Y.L. synthesized the photoalignment material.

## Additional information

**Competing interests:** The authors declare no competing financial interests.

