## [Peer Review File · Nature Communications]

Reviewers' comments:

Reviewer #1 (Remarks to the Author):

The authors demonstrate how a periodic twisted patterning of a surface in a twist nematic cell leads to a very stable freestanding array of $1/2$ twist disclinations. They also show how two such arrays can be superimposed. The fascinating study that demonstrates how to create and manipulate complex disclination networks will certainly attract the attention of researchers interested in anisotropic soft matter. Unfortunately, a discussion where further one can go with such disclination networks is missing. Further, the presentation is too much stressing the analytical modeling that is in fact not new. Therefore, the manuscript needs to be improved before I decide about my support of a publication in the Nature Communications. Below I list my particular remarks!

13: Chirality can also be a stabilizing factor what is the case in cholesterics and blue phases.

13, 21: It should be stressed that defect lines are twist disclinations.

16-17: Even closer are tightly confined blue phases discussed by Fukuda few years ago.

23: Cano-wedge needs at least a reference.

26: The Nature communications is a journal oriented toward broader audience therefore specific terms should be avoided or explained.

Fig 1: For the disclinations parallel and perpendicular to the director induced by the surface pattern show profiles as insets! Comment also handedness! It is not clear from where pictures c and d are coming! It would be good if they came from two orthogonal areas. The asymmetric variation of twist should be explained in the text.

27–35: Instead of listing unpublished, self-reference to justify Eqs.1 & 2, recent publications dealing with the similar analytical description of twisted disclinations (Afghah & Selinger, *ArXiv* 2017 and Copar et al. *PRE* 2016) should be included.

39-42: To justify the Eq. 3 for the interaction of two parallel disclinations at different heights one needs to stress that planar cell is used. Further, in the planar cell, the state with two opposite twist lines is metastable not equilibrium! The interaction of two parallel lines at the same height in a twisted cell should be discussed as well!

46- 59: The discussion of the interaction of nonparallel lines is confusing. In the extension to nonparallel cases it is not clearly stated in what kind of cell it is used. For connecting to the patterned situation, the twisted cell seems relevant. The question is whether one can treat short segments like long lines. What about intersections, etc.? When handedness comes in?

65: How strong is anchoring on treated surfaces?

74: Probably during sliding the non-patterned parts of two surfaces were in parallel cell configuration? It should be stated!

Fig 3e: The figure is not clear enough. Too tiny! What are 4 cylinders? In addition, the twist states of adjacent quadrants cannot be recognized!

102: It would be good to explain why diagonal lines are energetically better.

120-124: Analytical approaches are not new. They are useful for the understanding of main

features but are not enough for details like line crossing, line transitions, transient states, anisotropic elasticity, anchoring, etc.

125-128: Discussion about further evolution of the complex disclination arrays and possible use of the structures for applications is rather weak!

199: Reference 14 includes a misspelling!

Reviewer #2 (Remarks to the Author):

The paper Artificial web of disclination lines in nematic liquid crystals by Wang, Li and Yokoyama, reports a new experiment in which a free standing web of disclination lines in nematic liquid crystals is observed. The web shows robustness against deformation and reconfigurability.

While the claims of the paper seem very interesting and the videos certainly fascinating, at present I cannot make a decision about this paper, as there are too many experimental details that I cannot understand.

The main ones are the following:

- The authors claim (line 60) that they assembled a cell with two square substrates as those presented in fig. 1. However a few lines later (line 65) they say "...nematic LC was applied onto the patterned surface with a cover glass slide ...with uniform planar alignment". Therefore I cannot understand how the two surfaces of the cell are aligned, or if the authors are referring to two different cells, one with top and bottom patterned and one with only one glass patterned and one uniform planar.

- In line 74 the authors say "flipping one patterned substrate over the other we obtained a LC layer held between the square patterned substrates" and again I don't understand what this means. Are the authors using the samples described in the previous paragraph (if so, how?) or are they describing a different experiment here? How is the LC loaded in this case? (spin coated on both films and then merged, or just assembled a cell with two substrates and filled with LC)

- in line 79 the authors mention Fig.3 and the "rectangular region made of vertical stripes" at the center of the overlapped area. In the figure I can see a square region in the center, are the author referring to that?

- figure 3e puzzles me. The authors should improve the graphics and the explanation. It seems that it represents the two patterned substrates in which the alignment is everywhere shifted by 90 degrees. Is this a situation that the authors described in experiments? The disclination lines in this case seem to be running only from top to bottom cell wall. How is this reconciled with the previous claim and images of disclination lines running parallel to the walls of the cells?

In summary, I think that the paper has the potential to be interesting and publishable but in its present form it is not understandable enough to assess it. I would be happy to review the paper again if the authors accept to rewrite it and explain their experiment in more detail.

Reviewer #3 (Remarks to the Author):

I admit that I was skeptical first when I had read the abstract and had quickly overlooked the paper. It comes rather theoretical first, and the equations (1)-(3), which form the basis of the following interpretation of structures, come without references or derivation (the authors refer to a publication in preparation).

After careful reading the description of the exciting experiment and

understanding the potential of the method, I am very much in favor for a recommendation to publish this manuscript. The potential to use this surface patterning technique for the controlled preparation and study of disclination lines is remarkable. The defect lines can even be exploited to collect nanoparticles, to modulate transmitted light and to conduct interesting hydrodynamic experiments.

The only, more or less formal, criticism I have is that comprehensibility of the manuscript would benefit very much from a more detailed discussion of the derivation of the first three equations (for example in a supplementary information rather than in an unpublished manuscript). It is also very difficult to understand where the different regions in Fig. 3 a-d come from, it would be very helpful if the authors could add the surface alignment of the top and bottom plates in the four different regions (this could also be done in a supplementary information).

Itemized Responses to Reviewers' Comments:

First, we would like to thank very much all the reviewers for their constructive comments, pointing out a number of weak points of the manuscript, which we all agree. We have revised the manuscript according to the comments and address each point in such a way as to make the paper more comprehensible. We have also provided a detailed account of theoretical derivation of relevant formulas in *Supplementary Information*. Specific revisions are given below for your consideration.

Hiroshi Yokoyama

=====

Reviewer #1 (Remarks to the Author):

The authors demonstrate how a periodic twisted patterning of a surface in a twist nematic cell leads to a very stable freestanding array of 1/2 twist disclinations. They also show how two such arrays can be superimposed. The fascinating study that demonstrates how to create and manipulate complex disclination networks will certainly attract the attention of researchers interested in anisotropic soft matter. Unfortunately, a discussion where further one can go with such disclination networks is missing. Further, the presentation is too much stressing the analytical modeling that is in fact not new. Therefore, the manuscript needs to be improved before I decide about my support of a publication in the Nature Communications. Below I list my particular remarks!

13: Chirality can also be a stabilizing factor what is the case in cholesterics and blue phases.

As long as the pitch is not too short compared to the cell thickness so that the homogenous planar configuration is achievable as the ground state between a pair of uniform planar alignment substrates, the uniform twist can be superimposed on the present periodic structure without disrupting the disclination network. Except for the influence of anisotropy between K_{11} and K_{33} , which is unclear at present to the authors, the impact of chirality on the stability of the disclination web is neutral because of the linearity of the Euler-Lagrange equation for the azimuthal angle. A detailed account of this point has been given in *Supplementary Information* as a part of the derivation of the force expressions.

Short pitch cholesterics and blue phases seem likely to develop a 3D structure, breaking down the basis of the present study of planar structures. Empirical ubiquity of disclinations in short pitch cholesterics may suggest a enhanced possibility of constructing artificial web of disclinations, but we have not had any definitive evidence for such a possibility yet. Unfortunately, the present authors are not knowledgeable enough to include any conclusive statement about the effect of short pitch twist.

13, 21: It should be stressed that defect lines are twist disclinations. Specific mention of "twist" has been added in the text in order to emphasize that the disclinations are twist disclinations. It has also been stressed in line 23.

16-17: Even closer are tightly confined blue phases discussed by Fukuda few years ago.

The following reference has been added:

Fukuda, J. & Zumer, S. Novel Defect Structures in a Strongly Confined Liquid-Crystalline Blue Phase. *Phys. Rev. Lett.* 104, 017801(2010).

23: Cano-wedge needs at least a reference.

References to de Gennes' book (Ref.1) and

Cano, R. *Bull. Soc. Franc. Mine'ral. Cristallogr.* 91, 20–27(1968) have been added.

26: The Nature communications is a journal oriented toward broader audience therefore specific terms should be avoided or explained.

Following introductory paragraph has been inserted:

=====

For a given boundary condition at the bounding surface, the equilibrium distribution of the director can be theoretically found as the one that minimizes the Frank elastic free energy, written as a sum of contributions from the three independent mode of deformations referred to as the splay, twist and bend modes¹. These modes are associated with specific elastic constants, K_{11} , K_{22} and K_{33} , respectively. Here, we give only the salient results of the analysis, leaving the detailed derivation to *Supplementary Information*.

=====

Fig 1: For the disclinations parallel and perpendicular to the director induced by the surface pattern show profiles as insets! Comment also handedness! It is not clear from where pictures c and d are coming! It would be good if they came from two orthogonal areas. The asymmetric variation of twist should be explained in the text.

We have indicated clearly where **c** and **d** come from on **b**. Also, a detailed explanation about the handedness of the disclination lines in these pictures has been included as follows:

=====

The disclination lines in **c** and **d** are of the opposite handedness; in **c**, the left (right) side of the disclination is associated with $-\pi/2$ ($\pi/2$) twist from the bottom to the top surface, corresponding to the plus sign in Eq.(1) and hence is of the right handedness. That in **d** is of the left handedness. The oppositely twisted states across the disclination line give a different contrast under obliquely aligned polarizers.

=====

27–35: Instead of listing unpublished, self-reference to justify Eqs.1 & 2, recent publications dealing with the similar analytical description of twisted disclinations (Afghah & Selinger, *Axive* 2017 and Copar et al. *PRE* 2016) should be included.

Following two references have been added and the reference to authors' unpublished paper has been removed:

Čopar, S. et al. Sensing and tuning microfiber chirality with nematic chirogyral effect *Phys. Rev. E* 93, 032703(2016).

Afghah, S. & Selinger, J. V. Theory of helicoids and skyrmions in confined cholesteric liquid crystals *arXiv:1702.06896v1*, 22 Feb 2017.

We have also provided the detailed derivation of the formulas together with some discussions in Supplementary Information. With this description, we will not be publishing a separate paper on the theoretical formulation.

=====

Detailed description of the derivation of the theoretical formulas has been given in *Supplementary Information* instead of referring to a paper in preparation. This is also in response to a suggestion by Referee #3, and is expected to make the entire paper more complete and accessible to readers.

=====

39-42: To justify the Eq. 3 for the interaction of two parallel disclinations at different heights one needs to stress that planar cell is used. Further, in the planar cell, the state with two opposite twist lines is metastable not equilibrium! The interaction of two parallel lines at the same height in a twisted cell should be discussed as well!

The planar cell structure has been stressed by inserting the following sentence:

=====

It must be noted that the twist angles on both sides of disclinations are $\pm\pi$ or 0 depending on the same and the opposite handedness of the disclinations, so that the cell is of a uniform planar structure.

=====

A short discussion of the interaction force between disclination lines at the same height has been added as follows:

=====

Even when the parallel disclinations are laterally separated in the y direction, it is shown that the disclination lines of same (opposite) handedness repel (attract) each other. When, in particular, $d_1 = d_2$ and the lateral separation h is smaller than the cell thickness, the lateral force, both repulsive and attractive, is proportional to $1/h$ and is independent of the vertical position (for more details, see *Supplementary Information*).

=====

46- 59: The discussion of the interaction of nonparallel lines is confusing. In the extension to nonparallel cases it is not clearly stated in what kind of cell it is used. For connecting to the patterned situation, the twisted cell seems relevant. The question is whether one can treat short segments like long lines. What about intersections, etc.? When handedness comes in?

The description of this part has been improved to convey the configuration under consideration can be well understood. However, in order to prevent this theoretical part from becoming excessively long, the detailed derivation and explanation of the argument has been provided in the newly created *Supplementary Information*. Please refer to *Supplementary Information* for more details.

This paragraph has been revised as follows:

=====

The extra twist energy stored around the disclination lines is responsible for the repulsive and attractive forces. When a pair of straight disclination lines are intersecting at an oblique angle, the force between the disclination lines is localized in the vicinity of the intersection over the range of the cell thickness. Extending the analysis to the cases of obliquely oriented (non-parallel) straight disclination lines, we can show that the force between disclination lines (in the area of intersection) continuously varies between repulsive to attractive regime, proportional to $\cos\Phi$ with Φ being the angle between the disclinations. So, as one of the disclination lines of the same handedness is rotated in-plane relative to the other (under the fixed boundary), the repulsive force at the

intersection gradually decreases and vanishes when the lines are orthogonal. Further rotation leads to the increase of attractive force; in other words, the handedness of the disclination is switched to the opposite at $\Phi = \pi/2$ (see *Supplementary Information*).

=====

65: How strong is anchoring on treated surfaces?

The following sentence has been added:

=====

At saturated irradiation, the azimuthal anchoring energy of the photoaligned surface exceeds 10^{-4}mJ/m^2 , which can be considered perfectly strong for cells thicker than $1\mu\text{m}$.

=====

74: Probably during sliding the non-patterned parts of two surfaces were in parallel cell configuration? It should be stated!

Fig 3e: The figure is not clear enough. Too tiny! What are 4 cylinders? In addition, the twist states of adjacent quadrants cannot be recognized!

We depicted 4 cylinders to indicate the region of intersection, since the intersection points could be easily misunderstood in 3D perspective. We have completely recreated the figure to avoid confusions and to enhance visibility.

102: It would be good to explain why diagonal lines are energetically better.

Following explanation has been added in the text, and a more detailed account using some graphics has been provided in *Supplementary Information*:

=====

It is energetically more favourable for disclination lines to run periodically along the diagonal axes with $\pm \pi/2$ twists than shaping the square web, since the length of disclination line is reduced by a factor $1/\sqrt{2}$ and the super-twisted states near the $\pm\pi$ twisted quadrants in the square web are converted to the normal twisted states with the twist angle less than $\pi/2$ (for more details, see *Supplementary Information*).

=====

120-124: Analytical approaches are not new. They are useful for the understanding of main features but are not enough for details like line crossing, line transitions, transient states, anisotropic elasticity, anchoring, etc.

We agree that this part is intended only to lay out an analytical ground to illustrate the physical rationale for the main idea of the present work. Theoretical treatment of dynamical behaviors and the impact of anisotropic elasticity is much more complicated and beyond this simple analysis. We would like to leave them as open problems for future studies by theoreticians. Instead of citing a paper in preparation devoted to this analysis, *Supplementary Information* has been included in the present paper to provide sufficient information about the mathematical derivation to reach the equations given in the main text.

125-128: Discussion about further evolution of the complex disclination arrays and possible use of the structures for applications is rather weak!

Thank you for your comment. We have taken a liberty to enlarge the last paragraph of the paper to expand our view of its future potential as follows:

=====

Just like polymeric nanofibers, disclination lines in LCs carry tensile forces; but due to their structural origin, lacking in any physically distinct constituents, the disclination lines are superelastic and indefinitely extensible without yield point. If webs of condensed disclination lines can be fabricated by design, it would be foreseeable to develop disclination-integrated objects that exhibit a unique mechanical response to external fields and stimuli such as soft shape-memory. Moreover, the abrupt variation of anisotropic optical properties around disclination lines could also be potentially useful as optical media to manipulate the Pancharatnam-Berry geometric phase¹⁹ toward nanophotonic applications. Finally, it has also been demonstrated that disclination lines can trap a variety of nano- and micro-scopic objects for their higher energy states^{5,23}. It should be of interest to use the engineered disclination webs as template to assemble functional objects with nanometer scale precision, flexibility and responsivity. It may be particular intriguing if one can use the disclination web as tracks for cargo transport in small dimensions, much like cytoskeletons²⁴, taking advantage of the excitable property of liquid crystals. The stabilizing forces of disclination webs scale as the inverse of cell thickness and the pitch of the surface pattern. The disclination lines tend to behave more rigidly in smaller scales. Therefore, the finer the structure, the more robust the web becomes, making it an ideal platform for soft nanomachines.

=====

199: Reference 14 includes a misspelling!

Corrected: Biscarib -> Biscari

Thank you for pointing out this typographical error!

Reviewer #2 (Remarks to the Author):

The paper Artificial web of disclination lines in nematic liquid crystals by Wang, Li and Yokoyama, reports a new experiment in which a free standing web of disclination lines in nematic liquid crystals is observed. The web shows robustness against deformation and reconfigurability.

While the claims of the paper seem very interesting and the videos certainly fascinating, at present I cannot make a decision about this paper, as there are too many experimental details that I cannot understand.

The main ones are the following:

- The authors claim (line 60) that they assembled a cell with two square substrates as those presented in fig. 1. However a few lines later (line 65) they say "...nematic LC was applied onto the patterned surface with a cover glass slide ...with uniform planar alignment". Therefore I cannot understand how the two surfaces of the cell are aligned, or if the authors are referring to two different cells, one with top and bottom patterned and one with only one glass patterned and one uniform planar.

We are sorry for the confusing description. The last situation you are referring to is the real experimental conditions we used. This part of the text has been revised as:

=====

When temporarily covered for observation purposes with a glass slide whose surface was also photoaligned to give a uniform planar alignment, a regularly spaced array of 1/2-strength twist disclination loops was observed under the polarized optical microscope (Olympus) as shown in Fig. 1b.

=====

- In line 74 the authors say "flipping one patterned substrate over the other we obtained a LC layer held between the square patterned substrates" and again I don't understand what this means. Are the authors using the samples described in the previous paragraph (if so, how?) or are they describing a different experiment here? How is the LC loaded in this case? (spin coated on both films and then merged, or just assembled a cell with two substrates and filled with LC)

A new figure (Figure 3 in the revised manuscript) has been created to clearly show the fabrication procedure of this sliding cell. The new Fig.3 also includes a large area view of the cell so that the mutual relationship of different sections can be more readily understood.

The patterned substrates described in the previous paragraph were used. This part of the text has been revised to make this point clear as follows:

=====

Flipping one of the patterned substrates loaded with the nematic LC over the other (without the cover slide), we obtained a LC layer held between the square patterned substrates.

=====

- in line 79 the authors mention Fig.3 and the "rectangular region made of vertical stripes" at the center of the overlapped area. In the figure I can see a square region in the center, are the author referring to that? Yes, that is correct. In order to make it more readily understandable, we have revised the text as follows:

=====

At the centre of Fig.3a is a 45°-tilted rectangular region made by the overlap of the vertical stripes from the same quadrants of the square pattern on the top and bottom surfaces.

=====

- figure 3e puzzles me. The authors should improve the graphics and the explanation. It seems that it represents the two patterned substrates in which the alignment is everywhere shifted by 90 degrees. Is this a situation that the authors described in experiments? The disclination lines in this case seem to be running only from top to bottom cell wall. How is this reconciled with the previous claim and images of disclination lines running parallel to the walls of the cells?

We are sorry for the confusing artwork. We have carefully recreated the figure to be able to better convey the 3D configuration, and have made it Fig.5. In the original Fig.3e, the dark shaded cylinder running between the substrates were not the disclination lines, but were meant to highlight the region where the orthogonal disclination lines intersect. In this perspective view, we were constantly misled to see the wrong position of the intersection from the true intersection as viewed from the top. That is why we put that fictitious cylindrical shadow as a guide for eye, which turned out to be misleading.

In summary, I think that the paper has the potential to be interesting and publishable but in its present form it is not understandable enough to assess it. I would be happy to review the paper again if the authors accept to rewrite it and explain their experiment in more detail.

Reviewer #3 (Remarks to the Author):

I admit that I was skeptical first when I had read the abstract and had quickly overlooked the paper. It comes rather theoretical first, and the equations (1)-(3), which form the basis of the following interpretation of structures, come without references or derivation (the authors refer to a publication in preparation).

After careful reading the description of the exciting experiment and understanding the potential of the method, I am very much in favor for a recommendation to publish this manuscript. The potential to use this surface patterning technique for the controlled preparation and study of disclination lines is remarkable. The defect lines can even be exploited to collect nanoparticles, to modulate transmitted light and to conduct interesting hydrodynamic experiments.

The only, more or less formal, criticism I have is that comprehensibility of the manuscript would benefit very much from a more detailed discussion of the derivation of the first three equations (for example in a supplementary information rather than in an unpublished manuscript). It is also very difficult to understand where the different regions in Fig. 3 a-d come from, it would be very helpful if the authors could add the surface alignment of the top and bottom plates in the four different regions (this could also be done in a supplementary information).

Thank you very much for your patience to read through the paper despite your initial intuition. Following your comment, we have removed reference to unpublished work and added a sufficiently detailed account of the derivation of the equations as *Supplementary Information*.

We also have recreated some of the figures to enhance their comprehensibility and included detailed configurations of the cell in *Supplementary Information*.

REVIEWERS' COMMENTS:

Reviewer #1 (Remarks to the Author):

The authors have addressed all my remarks and have added an extensive supplement so that the manuscript is now substantially improved and can be published in the Nature Communications. Nevertheless, there is still a minor weak point that could be improved. The analytical description of the nematic ordering is assuming that director field is everywhere in the xy plane. That is the same as in Ref 13. This is certainly not always the case, nevertheless the Refs 14 and 15 where similar approach is in part used include also numerical simulations where this constrain is not needed. They show that for equal elastic constants, deviations are not pronounced and analytical approach provides a reasonably good insight in the director field structures. A comment on this should be added.

Reviewer #2 (Remarks to the Author):

In the paper "Artificial web disclination lines in nematic liquid crystals" by Wang, Li and Yokoyama, the authors describe a system consisting of a nematic cell with patterned substrate that forms a web of twist disclination lines. They provide a theoretical argument for the force between disclination lines, important for the stability of the disclination web, and they show a series of beautiful experiments.

In reference to my previous questions, I appreciate the incorporation of fig. 3a that clarified many of my doubts.

I think that this paper will fascinate the readers of Nature Communications and I recommend its publication with minor revisions.

A few additional questions and observations:

- the rounding of the corner of the disclination seems an interesting phenomenon, especially because from the images it is clear that the radius of curvature next to the corner is the same for all lines. Could the authors measure and comment on this characteristic curvature?
- in fig. 4 the authors describe in details various regions, highlighted in panel 4b (area with no disclinations, square pattern, bottom region with constant twist angle). I recommend that the authors identify with a letter, a symbol, or a color these different regions so that they can be easily recalled in the text.
- I personally still find fig. 5 confusing, especially fig. 5a, which in my opinion does not add much information (from fig. 1a deducing fig. 5a is straightforward). A 3D simulation of the director field would be much more useful, as the twist component would be evident.
- fig. 6 is referenced in the text as fig. 5 (line 139). Please fix this typo.
- From fig. 6 it appears that the disclinations always run at a distance of a few microns from the others. Based on this consideration, I feel very uncomfortable about the claim that the structure can be used for cargo like the cytoskeleton (line 161), if the lines never really intersect or touch. I would ask the authors to either remove that claim or justify it better.

Reviewer #3 (Remarks to the Author):

The authors have responded appropriately to the comments and criticism in the reports. I recommend publication of the paper.

Responses to Reviewers' Comments

=====

Reviewer #1 (Remarks to the Author):

The authors have addressed all my remarks and have added an extensive supplement so that the manuscript is now substantially improved and can be published in the Nature Communications. Nevertheless, there is still a minor weak point that could be improved. The analytical description of the nematic ordering is assuming that director field is everywhere in the xy plane. That is the same as in Ref 13. This is certainly not always the case, nevertheless the Refs 14 and 15 where similar approach is in part used include also numerical simulations where this constrain is not needed. They show that for equal elastic constants, deviations are not pronounced and analytical approach provides a reasonably good insight in the director field structures. A comment on this should be added.

To avoid further complications of the theory part in the main text, we have added the following remarks in Discussions and in Supplementary Note 1:

[Discussions]

The planar structure assumed here is not necessary valid particularly when the lateral deformation is significant over the length scale of the cell thickness and/or a short pitch chiral nematic LC is used instead of achiral nematic liquid crystals^{14,15}. It should, however, be worth mentioning that the out-of-plane orientation of the director can be efficiently suppressed even in these cases by using LCs with a negative dielectric anisotropy under a sufficiently intense electric field.

[Supplementary Note 1]

The limit of this planar assumption has been numerically examined in Refs.[14] & [15], confirming the validity in the present case of nematics.

=====

Reviewer #2 (Remarks to the Author):

In the paper "Artificial web disclination lines in nematic liquid crystals" by Wang, Li and Yokoyama, the authors describe a system consisting of a nematic cell with patterned substrate that forms a web of twist disclination lines. They provide a theoretical argument for the force between disclination lines, important for the stability of the disclination web, and they show a series of beautiful experiments.

In reference to my previous questions, I appreciate the incorporation of fig. 3a that clarified many of my doubts.

I think that this paper will fascinate the readers of Nature Communications and I recommend its

publication with minor revisions.

A few additional questions and observations:

- the rounding of the corner of the disclination seems an interesting phenomenon, especially because from the images it is clear that the radius of curvature next to the corner is the same for all lines. Could the authors measure and comment on this characteristic curvature?

This is certainly an interesting point on its own right. The rounding of disclination lines in response to unbalanced Frank elastic force and the line tension associated with the disclination has been discussed in Supplementary Notes through the Young-Laplace equation for the twist disclination line. The fact that the curvature of the disclination lines are the same at all corners is the consequence of this equation.

In the main text, we have added the following explanation:

... which is caused by the balance between the line tension of the disclination and the elastic stress generated by the imbalance of the twist deformations across the disclination line (for detailed discussions, see Supplementary Note 1 and Supplementary Note2).

A separate study is now underway in my group to elucidate the line tension of twist disclination from the curvature in a well-defined configuration of twist states. The results will be reported elsewhere shortly.

- in fig. 4 the authors describe in details various regions, highlighted in panel 4b (area with no disclinations, square pattern, bottom region with constant twist angle). I recommend that the authors identify with a letter, a symbol, or a color these different regions so that they can be easily recalled in the text.

While we understand the point and wish to be more specific, but the figures are already too crowded with added lines. We would like to keep the original photographs as much as possible at this stage.

- I personally still find fig. 5 confusing, especially fig. 5a, which in my opinion does not add much information (from fig. 1a deducing fig. 5a is straightforward). A 3D simulation of the director field would be much more useful, as the twist component would be evident.

We wish to leave Fig.5a as is for the benefit of readers who would find this figure helpful. It depicts additional information as regards the relative 3D positions of the orthogonally running disclination lines with respect to the surface orientation patterns. It also shows the distribution of director in the vertical direction at the intersection. These points are not readily evident from Fig.1a.

We tried to use a 3D simulated profile of the director, but the figure becomes so complicated that one can hardly recognize the twist structures anywhere, let alone the four segments about the intersection.

We apologize for this figure being useless for you, but at this stage, this seems to be the best compromise that we can create.

- fig. 6 is referenced in the text as fig. 5 (line 139). Please fix this typo.

Thank you for pointing this out. It has been corrected.

- From fig. 6 it appears that the disclinations always run at a distance of a few microns from the others. Based on this consideration, I feel very uncomfortable about the claim that the structure can be used for cargo like the cytoskeleton (line 161), if the lines never really intersect or touch. I would ask the authors to either remove that claim or justify it better.

Thank you for this significant comment. We have added the following cautionary statement, emphasizing the prematurity of this conjecture:

For this purpose, however, it remains to be a challenge to develop a scheme to transfer a cargo from one disclination to another without disrupting the integrity of the disclination lines.

=====
Reviewer #3 (Remarks to the Author):

The authors have responded appropriately to the comments and criticism in the reports. I recommend publication of the paper.